# Apathy in Parkinson’s Disease: Defining the Park Apathy Subtype

**DOI:** 10.3390/brainsci12070923

**Published:** 2022-07-14

**Authors:** Ségolène De Waele, Patrick Cras, David Crosiers

**Affiliations:** 1Translational Neurosciences, Born-Bunge Institute, Faculty of Medicine and Health Sciences, University of Antwerp, 2650 Edegem, Belgium; patrick.cras@uza.be (P.C.); david.crosiers@uza.be (D.C.); 2Department of Neurology, Antwerp University Hospital, 2650 Edegem, Belgium

**Keywords:** Parkinson’s disease, apathy, neuropsychiatry, non-motor subtyping

## Abstract

Apathy is a neurobehavioural symptom affecting Parkinson’s disease patients of all disease stages. Apathy seems to be associated with a specific underlying non-motor disease subtype and reflects dysfunction of separate neural networks with distinct neurotransmitter systems. Due to the complicated neuropsychiatric aetiology of apathy, clinical assessment of this invalidating non-motor symptom remains challenging. We aim to summarize the current findings on apathy in Parkinson’s disease and highlight knowledge gaps. We will discuss the prevalence rates across the different disease stages and suggest screening tools for clinically relevant apathetic symptoms. We will approach the fundamental knowledge on the neural networks implicated in apathy in a practical manner and formulate recommendations on patient-tailored treatment. We will discuss the Park apathy phenotype in detail, shedding light on different clinical manifestations and implications for prognosis. With this review, we strive to distil the vast available theoretical knowledge into a clinical and patient-oriented perspective.

## 1. Introduction

Non-motor subtyping in Parkinson’s disease (PD) has garnered increasing interest in the past few years. While useful, motor subtyping does not adequately portray PD’s highly heterogeneous clinical presentations, as motor symptoms change during the disease course. Non-motor subtyping may prove a valid and more precise alternative, allowing for a patient-tailored approach and treatment [1,2]. Current findings point towards apathy as a distinct marker of a non-motor disease subtype: the Park Apathy subtype [3,4]. The manifestation of apathy within the non-motor spectrum of PD was first described in 1982 by Dr. Rabin. In his case series of 13 patients, he described that ‘apathy is also common […] this can be the most debilitating symptom’ [5]. In the past, it was considered a late-stage symptom occurring predominantly in elderly patients [6]. Research has shown however that apathy manifests itself in all disease stages, and may serve as a prodromal symptom in some [6,7,8,9]. Presence of apathy in PD patients has been linked to increased motor burden, reduced quality of life (QoL) and has been identified as a risk factor for motor complications and cognitive decline [10,11,12,13,14]. Despite the profound implications, screening for apathy is generally not included in daily clinical practice. Patients and their loved ones may struggle to pinpoint the underlying problem, attributing the symptoms to fatigue or unwillingness on the patient’s part [15,16]. Medical professionals as well struggle to identify it during routine follow-up and often apathetic patients are thought to suffer from depression or cognitive decline [16]. Trials have been undertaken to identify treatment options but successful results are sparse [17]. This is in part due to the one-dimensional approach undertaken in most trials. Apathy is usually described as a neuropsychiatric symptom, but it is more accurate to consider apathy as a behavioural state: the quantitative reduction of self-generated voluntary and purposeful behaviours […] [18]. This altered behavioural state can arise from dysfunction in different neural networks, regulated by specific neurotransmitters, manifesting itself clinically into separate syndromes [18]. These symptoms may present themselves separately or in combination, requiring a customized approach.

This review aims to summarize the current knowledge about apathy in PD in the different disease stages. Subsequently, we will approach the underlying psychological and neural concepts and the pathophysiology practically, including possible helpful imaging biomarkers. Lastly, we detail the phenotype of the Park Apathy subtype and discuss potential treatment options.

## 2. Materials and Methods

We collected articles published between 1 January 2014 and 1 December 2021 by searching the following databases: Pubmed, Web of Science, and Google Scholar. We used the following search terms: ‘apathy’, ‘neuropsychiatric symptoms’, ‘motivation’, ‘Parkinson’, ‘Parkinson’s disease’, ‘imaging’, ‘pathophysiology’, ‘prodromal’, ‘prevalence’, and ‘treatment’, or a combination thereof. The abstracts of the resulting articles were scanned and only those relevant to the scope of this review article were included. References of the included articles were browsed and pertinent papers were included after analysing their respective abstracts. 

## 3. Prevalence of Apathy during the Disease Course

Epidemiological studies show a wide range of prevalence rates of apathy in PD (see Table 1) [4,11,14,19,20,21,22,23,24,25,26,27,28,29,30,31,32,33,34,35,36,37,38,39,40,41]. These discrepancies can be attributed to several factors.

First, methodologies to determine apathy vary significantly across the studies. Several scales are available to the clinician to evaluate apathy. Symptoms can be rated by the patients, their caregivers, or the clinician. The assessment tools range from quoting one item on an non-motor symptoms (NMS) scale to apathy-specific scales, which are quoted numerically or through a Likert scale [42]. Despite wide and frequent use, many scales have poor sensitivity or are less appropriate to evaluate apathy in specific disease populations. The Movement Disorders Society Unified Parkinson’s Disease Rating Scale (MDS-UPDRS) part I is a frequently used scale to assess NMS. Yet, it performs poorly compared to apathy-specific scales such as the Apathy Scale (AS) or the Lille Apathy Rating Scale (LARS). Compared to the LARS, the UPDRS apathy item has a sensitivity of only 33% [31,43]. The Geriatric Depression Scale-15 (GDS-15) is used in the Parkinson’s Progression Markers Initiative (PPMI) cohort (ClinicalTrials.gov Identifier: NCT01141023). Some studies have used a subscore of the GDS-15 to identify apathy [44,45]. However, this subscore is unsuited for use in de novo PD patients [46]. The MDS Task Force on rating scales for PD recommends the use of the UPDRS item 1.5, AS or LARS [47,48]. The UPDRS item 1.5 must be considered a screening tool, due to its apparent limitations [48]. The LARS scale was not granted the classification ‘recommended’ by the Task Force in 2008, but now fulfils the necessary criteria for recommendation following further research [47]. The LARS has an additional benefit, allowing for differentiation of subtypes of apathy, which we discuss further below [49].

Second, apathy can reflect an underlying mood disorder, a confounding factor that is not consistently excluded. Current apathy scales may not be sufficiently refined to detect these subtleties, and a separate evaluation of concurrent depression is recommended [47]. Third, apathy is common in the elderly without underlying neurological or psychiatric conditions. Apathy in community-dwelling adults is estimated to be about 11–29.4%, and the prevalence increases with age and functional decline [50,51]. Lastly, apathy in a patient can fluctuate during disease progression, which could either be attributed to the introduction of medication or the symptom’s natural course [14,36,52]. Few studies have undertaken longitudinal follow-up of apathy or have monitored the evolution of this symptom at the individual patient level [14,19]. Martin and colleagues mapped the development of apathy scores per patient for two years, yet no clear pattern emerged [52].

Considering these confounding factors, current data suggest that apathy in PD is present in all disease stages. It can even manifest in the prodromal phase, e.g., PD patients start traveling less almost 8 years before diagnosis, in the absence of overt functional decline, mood disorders or motor deficits at that time [53]. In retrospective studies, 14.3–31% of PD patients reported decreased initiative 2–9 years predating their diagnosis [2,54,55]. Recall bias is, however, inherent to retrospective studies; therefore, evaluating individuals at high risk of developing PD may offer a more objective insight into the prodromal stage. Patients suffering from REM sleep behaviour disorder (RBD) have a 34–73% chance of converting to a clinically manifest synucleinopathy within five to ten years [56,57,58]. Apathy is common in these patients, affecting around 46% [43]. Following conversion to manifest PD, apathy remains more prevalent in patients with RBD than those without RBD [59,60]. An additional common prodromal PD sign is hyposmia; idiopathic anosmia has a lower PD conversion rate than RBD, with reports of 10% at ten years [61]. Apathy may be more prevalent in hyposmic PD patients, possibly associated with a higher odour threshold and decreased discrimination and identification of odours [8,62,63,64]. It was not related to subjective changes in smell [4]. 

The decline in the prevalence of apathy in the first few years after PD diagnosis has been attributed to the introduction of dopaminergic medication [19,30,36], which was corroborated by several studies [19,30,36]. Prevalence rates during the disease course vary (see Table 1). In de novo, treatment-naive PD patients, reported rates vary from 15% to 40.8% [4,11,20,30,37,38,39,41]. Patient cohorts around 1–1.5 year disease duration show similar prevalence rates: 18.6 to 48.3% [21,22,23,39]. At 2–4 years after diagnosis, the proportion of apathetic patients drops to 20.2–28%, possibly reflecting an increased dosage of dopaminergic replacement treatment (DRT) [20,23,65]. Further progression of the disease accompanies a further increase in the number of apathetic patients. [23] Reported prevalence rates vary from 14.7–72% in patients with 5–10 years of disease duration, [24,25,26,27,28,29,31] to 13.9–63% after more than 10 years of disease [14,32,34,35]. Longitudinal follow-up studies with sensitive apathy-specific scales can shed more light on this complex matter. 

**Table 1 brainsci-12-00923-t001:** Prevalence rates of apathy during the disease course. * Marks studies in which underlying depression was excluded.

Author (Year)	Rate of Apathy (%)	Disease Stage (Mean ± SD (Years))	Measuring Tool
** Prodromal stage **			
Pont-Sunyer et al. (2015) [2]	50%	−10–2 years	Patient perception
Gaenslen et al. (2011) [54]	23.7%	−8.8 years	Patient perception
Darweesh et al. (2017) [53]	NA (case-control)	−7.7 years	IADL–traveling subscore
Durcan et al. (2019) [55]	14.3%	−2.1–0.7 years	NMSQuest
** High-risk populations **			
Barber et al. (2018) [43]	46%	RBD	LARS
** De novo, untreated **			
De La Riva et al. (2014) [39]	16.7%	±0.5	UPDRS
Hinkle et al. (2021) [11]	16.9%	0.5 ± 0.5	UPDRS
Liu et al. (2017) [37]	17.29% *	1.26 ± 1.25	LARS
Dujardin et al. (2014) [4]	13.7% *	1.3 ± 0.9	LARS
Oh et al. (2021) [41]	30.1%	1.6 ± 1.9	NPI
Santangelo et al. (2015) [30]	33.3%	<2	Diagnostic criteria [66]
Cho et al. (2018) [38]	58.8%	2.1 ± 1.97	NMSS
Leiknes et al. (2010) [20]	29.1%	2.3 [0.4–10]	NPI
** Early disease stage **			
Benito-León et al. (2012) [21]	21.7% *	1.3 ± 0.6	LARS
Cubo et al. (2012) [22]	33.4%	1.3 ± 0.6	LARS
Ou et al. (2021) [23]	18.6%	1.5	LARS
De La Riva (2014) [39]	30.2%	±3	UPDRS
** Mild disease **			
Isella et al. (2002) [25]	43.3%	4.9 ± 4.4	AES-S
Eglit et al. (2021) [29]	71.7%	5.5 ± 5.2	AS
Kulisevsky et al. (2008) [26]	48.3%	5.65 ± 4.94	NPI
Lieberman et al. (2006) [28]	44% *	6.2 ± 5.9	NPI
Oguru et al. (2010) [27]	17% *	6.3 ± 4.4	AS
Kirsch-Darrow et al. (2006) [24]	28.8% *	6.4 ± 5.7	AES
Butterfield et al. (2010) [33]	14.7% *	7.07 ± 4.96	AES-S
Kirsch-Darrow et al. (2009) [31]	31.4% *	8.1 ± 5.9	AS
** Advanced disease **			
Aarsland et al. (1999) [34]	16.5% 4.3% *	12.6 ± 5.1	NPI
Stella et al. (2009) [32]	38%	12.7 ± 6.2	NPI
Pedersen et al. (2009) [14]	13.9%	13.0 ± 4.7	NPI

Abbreviations: SD: standard deviation; NA: not applicable; IADL Lawton Instrumental Activities of Daily Living; NMSQuest: Non-Motor Symptom Questionnaire; RBD: REM sleep behaviour disorder; LARS: Lille Apathy Rating Scale; NPI: Neuropsychiatric Inventory; NMSS: Non-Motor Symptom Scale; UPDRS: Unified Parkinson’s Disease Rating Scale; AES-S: Apathy Evaluation Scale–Self-rated; AS: Apathy Scale.

## 4. Pathophysiology

### 4.1. Psychological Model

To understand the pathophysiology of apathy it is necessary to evaluate how apathy arises as a behavioural concept. Apathy is essentially the reduction of voluntary, goal-directed behaviour (GDB) [18]. The neuro-cognitive formulation of how GDB arises is convoluted, so we provide an abbreviated model based on Brown and Pluck’s theory (Figure 1) [67]. The process of GDB emerges from an interaction of internal and external drives, intention, planning, motivation, and emotional state. Theoretically, interference in any of these processes can lead to apathy [18,67].

Stuss and coworkers described three distinct subtypes of apathy. First, difficulties with self-activating thoughts or initiating the necessary motor functions for GDB are the predominant feature in the first type. We summarized this as a reduction of ‘internal drive’. This reduced internal drive and response starkly contrast the preserved reaction to external drives and stimuli [18,68]. In an everyday setting, these patients do little unless instructed [69,70]. Daily productivity is low and there is no variation in activities in daily life [49], resulting in severe inertia that can be reversed successfully with external stimuli [18]. This subtype of apathy is sometimes referred to as ‘behavioural apathy’; however since all forms of apathy lead to a reduction in GDB, we suggest an alternative terminology [71]. Aphrenic apathy, derived from the Greek word for ‘inability to think’, is a more apt description of this subtype. 

A disruption in the planning aspect of GDB leads to ‘cognitive’ apathy or cognitive inertia. Faulty executive processing lies at the basis here, rendering difficulties with planning, working memory, rule-finding, and set-shifting [18]. The underlying executive dysfunction makes it difficult to plan the actions needed to perform GDB [18,67,68]. Executive dysfunction does not always reflect underlying dementia but may herald it [72]. This clinically manifests as a decreased interest, or more accurately, a decreased (intellectual) curiosity [69]. These patients spend little or no time on leisure activities and have few interests. Often, they do not wish to pursue (new) hobbies or social engagements. They quickly give up on a task when facing difficulties, reflecting their executive dysfunction [49,69]. 

A final subtype of apathy is an underlying reward deficiency syndrome, in which a patient cannot relate the GDB to the (pleasurable) outcome or reward. This is the result of emotional blunting or reduced emotional resonance [18,68,69]. This third subtype is often referred to as ‘motivational’ apathy [18,68]. It results in a reduced emotional response, for instance when the patient is confronted with upsetting news or watching something humorous. Patients can also display a decreased concern for their families and often no longer inquire after their health and well-being spontaneously [18,49].

It is necessary to differentiate this apathy subtype from the symptoms of an underlying depression or mood disorder. Apathy can also be related to anhedonia, resulting in decreased GDB [67,73]. Apathy in patients suffering from a depressive episode can improve with adequate treatment of their mood disorder [74].

Aside from these three widely accepted apathy subtypes, a fourth dimension called ‘self-awareness’ was initially proposed by Stuss and coworkers [68]. These authors described self-awareness as a critical component of GDB. They defined it as ‘[…] a metacognitive ability, necessary to mediate information from a personal, social past and current history with projections to the future […]’ [68]. The LARS was developed with this fourth dimension in mind, reflected by a fourth and independent cluster in their data analysis, separate from depression [49]. The question remains whether reduced self-awareness can be considered an underlying mechanism of apathy or a different construct altogether [69]. Self-awareness in essence organizes an individual’s understanding of a social environment and the function of this individual within it [75]. Clinically, impaired self-awareness can manifest as anosognosia, or reduced insight into one’s own physical limitations due to an illness. Reduced self-awareness has often been described in PD patients and is associated with cognitive decline [76,77,78]. Clinically, this may manifest itself in social interactions, where the patient might be quite headstrong in an argument, unwilling to concede to another’s point of view. This results from a decrease in self-reflection, making it difficult to assess one’s own faults accurately [49].

### 4.2. Neural Networks

It is often assumed that apathy results from a pure hypodopaminergic state, as it often can arise following dopamine withdrawal for Deep Brain Stimulation (DBS) surgery. Dopaminergic treatment has shown improvement in some patients, but a more complex model is required to explain the implicated neurotransmitter systems [19]. Despite adequate dopaminergic treatment, apathy occurs in PD patients, and the severity of apathy is independent of medication dosage [39,79]. Apathy can co-occur in PD patients suffering from impulse control disorders related to a hyperdopaminergic state [80,81]. Co-occurrence of apathy and impulse disorders was also reported in other neurological disorders [82]. Lastly, animal models and imaging studies in patients have shown the involvement of other neurotransmitter systems [83,84,85]. To further study the complex underlying physiology of apathy, the definition of the neuroanatomical correlates is an important starting point. Generally, the occurrence of apathy can be reduced to a dysfunctional circuitry between the frontal lobes and the basal ganglia. Within this circuitry, separate networks can be identified (see Figure 2) [68,86,87]. 

‘Behavioural’ or ‘aphrenic’ apathy is often equated to a lack of initiation or internal drive to perform the GDB [18,68]. It is often referred to as an auto-activation deficit, with a reduced response to internal stimuli [18]. This type of apathy is often considered the most severe form. It has been described in bilateral dysfunction of the pathway between the dorsomedial prefrontal cortex (dmPFC) and anterior cingulate cortex (ACC) with the dorsal striatum, paramedian thalamus and the internal part of the globus pallidus (GPi) [88,89,90,91,92,93,94,95,96]. Similar syndromes have been described in uni- or bilateral lesions of the supplementary motor area (SMA) [70]. These are regions of interest in the ‘lateral orbitofrontal cortex’ circuit as described by Garrett and colleagues, which also receives input from temporal gryi and projects to the substantia nigra pars reticulata [87]. This circuit is partially dopamine-mediated, as evidenced by reduced dopaminergic binding and response to DRT [88,89,92,97]. 

Executive dysfunction leads to ‘cognitive’ apathy, where planning difficulties interfere with GDB. The dorsolateral PFC, cooperating with the ACC, is vital to executive processing, resulting in apathetic behaviour when lesioned [18,87,98,99]. This region has projections to the lateral parts of the dorsal striatum [87,92,100,101,102]. The lateral dorsal striatum also receives input from the posterior parietal cortex [87]. Cognitive apathy has been linked to decreased functional connectivity (FC) between the orbitofrontal cortex (OFC) and the right putamen [103]. We assume that the ‘dorsolateral PFC’ circuit is largely acetylcholine-mediated due to its implication in executive dysfunction [104]. In Alzheimer’s disease patients with predominant cognitive apathy, there was reduced response to dextroamphetamine administration, suggesting some possible dopaminergic involvement as well [105].

‘Motivational’ apathy is mediated by the mesocorticolimbic pathway or the reward system [18,68,106]. Involved regions are the orbitomedial PFC, the ACC, the ventral striatum, the ventral pallidum and the dopaminergic midbrain neurons [18,107]. This system is mediated by the amygdala, hippocampus, thalamus, lateral habenular nucleus, the dorsal PFC, as well as the pedunculopontine nucleus and raphe nucleus in the brainstem [107,108]. Patients with ‘motivational’ apathy according to the LARS showed altered FC between the left inferior frontal gyrus and the left pallidum. There was an increased FC between the left inferior frontal gyrus and the right caudate [103]. Apathetic PD patients showed selective impairment of reward processing, reducing their ability to differentiate between favourable and unfavourable outcomes [109]. Dopamine plays an important role in this circuit, yet its relation to manifest apathy is complex [110]. Administration of dopamine agonists blunts reward sensitivity in healthy adults, while use in apathetic patients shows promise as a potential therapy [111,112]. Yet, some studies found no difference in dopaminergic uptake between apathetic and non-apathetic patients [84]. Serotonin could act as a modulator, with reduced uptake found in critical parts of the mesocorticolimbic pathway in apathetic PD patients [84,85]. The uptake reduction was proportional to apathy’s severity [84]. Reduced serotonergic uptake in the raphe nucleus was also associated with the presence and severity of apathy in possible prodromal PD patients [113]. 

In our abbreviated and modified model in Figure 1, we propose a new role for self-awareness in developing GDB. Recent imaging findings have identified common underlying brain regions and networks in patients with reduced self-awareness and apathy. The precuneus is part of the default-mode network and plays an important role in self-awareness [114,115,116]. Studies found that isolated apathy in PD was associated with atrophy and hypometabolism of the precuneus compared to healthy controls [117,118]. Other regions of interest in self-awareness are the ACC, the posterior cingulate cortex (PCC), the temporoparietal junction, the ventromedial and dorsolateral prefrontal cortex and the insula [119,120,121]. 

It is unlikely that a different type of apathy develops in each patient. Patients with typical auto-activation deficit lesions were also shown to have reduced reward sensitivity [97,122]. One study found that apathy profiles differed, depending on disease stage. In stable PD patients, defined by the authors as well-controlled motor symptoms without fluctuations and absence of dementia, there was a trend towards decreased intellectual curiosity or ‘cognitive’ apathy. In PD patients with motor fluctuations without dementia, mostly intellectual curiosity and action initiation were inhibited. In PDD, both domains as well as self-awareness were decreased. Interestingly, in all groups, motivational apathy, as measured by the emotion subscore of the LARS, did not differ significantly from healthy controls [13]. Another study found a predominant decrease in intellectual curiosity in early-stage PD patients [4]. These findings suggest that apathy subtypes might have a distinct temporal profile. 

### 4.3. Imaging Biomarkers

Aside from neurotransmitter changes in different networks, additional imaging biomarkers of apathy have been investigated. Changes have been reported in grey-matter volume (GMV), white and grey matter integrity, FC and network analysis, regional homogeneity (ReHo), glucose metabolism, and resting activity pattern.

Decrease in GMV in the subgenual AAC, left superior temporal, left precuneus, inferior parietal, right superior frontal, and the dorsolateral part of the caput of the left caudate nucleus is related to the presence of apathy. Severity of apathy was related to morphological abnormalities in the superior cerebellar peduncle decussation, bilateral posterior cerebellum and vermis, left superior frontal gyrus, and left nucleus accumbens [84,117,123]. GMV increases were noted in the left superior frontal gyrus and cerebellar vermis [117]. Other imaging studies, however, could not confirm these changes [86,124,125]. The connectivity between the parietal cortex and frontal lobes might explain part of these findings, as frontal lesions lead to parietal hyperactivity [116]. Input of temporal gyri has also been described in the ‘lateral orbitofrontal cortex’ circuit of the basal ganglia, implicated in ‘behavioural’ apathy [87].

Fractional anisotropy (FA) was significantly decreased in the genu and body of the corpus callosum, bilaterally in the anterior corona radiata and the left superior part of the corona radiata and left cingulum in apathetic PD patients. The grade of integrity was related to apathy severity [126]. Another study found reduced FA in the anterior thalamic fibres, the cingulate bundle, and the corpus callosum’s interhemispheric connections and projection fibres. FA was also decreased bilaterally in the medial thalamus [84].

FC was reduced between the left ventral striatum and left frontal lobe in apathetic PD patients. Reduced FC between ventral and dorsal striatum and left frontal lobe, between the limbic region of the left frontal lobe and left striatum, between the caudal and rostral frontal lobe and right striatum and in between subdivisions of the left frontal lobe was related to increased severity of apathy [86] A regional network analysis could not find differences in connectivity between apathetic and non-apathetic PD patients [124]. 

Analysis of low-frequency function (ALFF), which evaluates the resting state of the entire brain, showed decreased ALFF signal in the left supplementary motor region, left inferior parietal love, left fusiform gyrus, and bilaterally in the cerebellum [127]. 

ReHo measures synchronization of local neural activity. In apathetic PD patients, ReHo was decreased in right caudate and dorsal ACC [128]. Some studies found reduced glucose metabolism in the precuneus bilaterally and right lingual gyrus and increased metabolism in the middle frontal gyrus in apathetic patients [117]. Additionally, the severity of white matter hyperintensities on FLAIR sequence also showed a link to apathy in PD, independent of depression [129]. These findings suggest top-down control from other cortical regions and support the involvement of the parietal cortex in certain subtypes [87,116]. 

## 5. Park Apathy 

The Park apathy subtype has been associated with more severe motor symptoms, confirmed in observational studies [3]. Apathetic patients score higher on the UPDRS motor scale than their non-apathetic peers, excluding confounding factors such as disease duration or age [8,9,22,130,131]. This difference is already manifest at diagnosis, before the introduction of DRT [4,7,8,22,37]. As discussed above, apathy may fluctuate during the disease course [14,19]. Persisting apathy was linked to a more significant increase in motor symptoms during a four-year follow-up period compared to those with incidental apathy [14]. Severity may also play a role, as the grade of motor disability and apathy go hand-in-hand in specific cohorts [4,7,23,37]. Despite extensive research, not all research groups found increased motor severity in this group [13]. The discrepancy may be explained by the effect of persistent and incidental apathy [14].

Specific motor symptoms in apathetic patients differ as well. They have increased body sway in a quiet stance compared to non-apathetic patients, reflecting a more pronounced underlying postural instability [132]. Even in early PD, axial symptoms are more frequent and pronounced in this group [8,22,37,133]. Freezing of gait in ON state is linked to a higher grade of apathy and was less responsive to medication in this group [134]. They are at increased risk of developing motor complications such as fluctuations and dyskinesias earlier in the disease course [11]. However, the relation of apathy to motor fluctuations may be more complex, as dyskinesias at baseline were a predictive factor of worsening apathy [36]. Apathy is possibly more common in patients with right-sided PD onset, and patients with left-sided onset had decreased odds of developing apathetic behaviour [22,135]. 

Apathetic patients also suffer from more NMS, evidenced by higher scores on non-motor scales at disease initiation [37]. Symptoms such as anhedonia, sleeping difficulties and fatigue occur more often [4]. Increased fatigue has quite consistently been linked to apathy [4,64,136,137]. Especially ‘motivational’ apathy is a predictor of worsening fatigue in early PD [138]. On fatigue-specific scales, apathetic patients primarily report mental fatigue related to decreased intellectual curiosity [136]. 

The relationship between apathy and depression is complex. Apathy can arise as a symptom of an underlying depression but can also manifest as a distinct symptom altogether. In most cohorts, apathetic patients score higher on depression scales in early and advanced PD [7,8,13,14,23,130], whereas in another sample depressive symptoms were not noted [139]. Possibly higher depression scores are a risk factor for developing apathy [14,36]. There is a possible overlap between the assessment scales for depression and apathy, and a separate evaluation of both is still recommended [49].

Early on, apathetic patients generally perform normally on basic cognitive screening tests [4,8,140]. Executive dysfunction and memory deficits, however, do become apparent when an extensive neuropsychological battery is performed. These patients display mild executive dysfunction, evidenced by lower scores on the interference task of the Stroop test, the Benton Judgment of Line Orientation Test, and the Letter Fluency test [30,125,140]. These impairments become more conspicuous over the years [9,14,141]. In more advanced stages, persistent apathy was linked to greater global cognitive decline [14]. Dujardin and coworkers found preserved cognitive efficiency in advanced apathetic PD patients without dementia. Attention, working memory, executive functions, language, and visuospatial skills were significantly decreased nonetheless [142]. Others report similar declines in executive function and visuospatial abilities in the absence of dementia [33,139,141,143]. 

Apathy is a possible risk factor for the development of dementia. Prevalence of PD with minimal cognitive impairment (PD-MCI) increases with disease progression, but conversion to either normal cognition or PD dementia (PDD) is possible [144]. Combined with the fluctuating nature of apathy, this may impede forming robust conclusions [19]. It is generally assumed that apathy is more frequent in both PD-MCI and PDD [77,145]. In a longitudinal analysis, self-rated apathy scores were linked to current and future cognitive scores, but were not predictive of conversion to PDD [52]. A longitudinal study found that those with both incident and persistent apathy had a more significant decrease in cognitive functions after four years, with a more pronounced change in those with persistent apathy. The rate of dementia in the persistent apathy group did not significantly differ from the baseline [14]. In another sample, the conversion rate to dementia was higher in apathetic patients. They also noted decreased cognitive scores at baseline, but a much more pronounced reduction in scores at follow-up in the apathetic group [9].

The above suggests that apathy is a marker of a more severe disease phenotype, with a higher motor and non-motor burden. Subtyping based on the presence of apathy has yet to be applied in large cohorts, but current evidence shows promise [146]. Evidence suggests that persistent apathy may be a more significant risk factor than purely incidental apathy [14].

## 6. Treatment

Despite quality research on the topic, effective treatment for apathy in PD is lacking [17]. This is partly due to the variety of screening methods and follow-up duration as discussed above. The one-size-fits-all approach complicates matters further. Patients may suffer from different subtypes or combinations thereof, requiring customized treatment. Screening for and identifying the dominant subtype(s) per patient might be helpful in future research, allowing for a more patient-oriented approach. We highlight the most promising strategies below. For a more extensive overview, we would like to direct the reader to a review article that delves deeper into the subject [17]. 

### 6.1. Pharmacological

As discussed above, many neurotransmitters are involved in the underlying process of apathy. Evidence has been found of dopaminergic, serotonergic, and cholinergic involvement [79,83,88,89,92,97].

Use of DRT has shown promising results, and administration of dopamine agonists is often most successful. A recent meta-analysis concluded that using rotigotine improved apathy scores, which was not confirmed in a more recent placebo-controlled study [147,148]. Other dopamine agonists such as pramipexole or apomorphine might also be beneficial, as patients score better on the items ‘intellectual curiosity’ and ‘self-awareness’ after administration [112,149]. Global apathy scores improved in those receiving apomorphine when combined with rotigotine [149,150]. Rotigotine and pramipexol were effective in reversing an auto-activation deficit in a case series [151]. 

DRT is assumed to improve apathy in the long term, as evidenced by the decreasing prevalence after the introduction of medication [23]. Apathy scores do not differ significantly in ON or OFF stages, showing no significant response to DRT in the acute phase [152]. No studies comparing different DRT strategies in these patients are available.

Results on serotonergic treatment are scarce. Selective serotonin receptor inhibitors are known to induce flat affect and apathy, both in healthy individuals and PD patients [153]. A cross-over study in 25 PD subjects with 5-hydroxytryptophan, a precursor of serotonin, had no significant impact on apathy scores [154]. Use of both selective serotonin and serotonin noradrenaline reuptake inhibitors (SSRI and SNRI respectively) did not significantly alter apathy scores compared to baselines [155].

In those already receiving optimized dopaminergic treatment without PDD, add-on of rivastigmine improved apathy scores [79]. Although rivastigmine was reported to improve apathy in PDD in a few case reports, a more extensive patient series showed no improvement in this group [104,156]. Use of rivastigmine decreased caregiver distress associated with apathy [157]. Galantamine might be effective in apathetic PDD patients [158].

Other strategies include the use of stimulants. Administration of dextroamphetamine in a PD sample with cognitive decline improved apathy scores in nearly a third of patients. Most of these patients were already receiving cholinesterase inhibitors [159]. Singular positive reports have been published on the use of methylphenidate, istradefylline, MAO inhibitors, yokukansan, and exenatide [160,161,162,163,164,165]. Bupropion and choline alphoscerate, a cholinergic precursor, was shown to be effective in treating apathy in other neurodegenerative diseases [166,167,168]. A case report of a patient suffering from an auto-activation deficit reported spectacular improvement of symptoms following administration of tricyclic antidepressants [169]. 

### 6.2. Non-Pharmacological

Non-pharmacological treatment options have garnered increasing interest. Exercise especially is beneficial in the treatment of both motor and NMS [170,171]. For the treatment of apathy, exercise and physical activity may also prove useful. A longitudinal study found that patients with baseline higher activity levels had improved apathy scores at follow-up. Apathy scores at baseline were not related to activity level [172]. Others however found very little difference between those following an intensive exercise schedule with sessions thrice a week and those without intervention. Only those following individual therapy showed slight improvement [173]. Apathy scores did improve in patients following biweekly Nordic walking sessions over 12 weeks, compared to control patients [174]. Though there is some evidence for a positive effect of dance, a recent meta-analysis concluded it was not superior to self-directed exercise or the best medical treatment [175,176]. There is need for structured research into the matter, wherein different physical activities and interventions are systematically researched and compared. As current evidence does not support one type of physical activity above another, it is advised to tailor the type of physical exercise to the patients’ needs and preferences [177].

A small body of evidence exists for using repetitive transcranial magnetic stimulation (rTMS). A cross-over study found that rTMS over the supplementary motor area improved apathy scores compared to placebo [178]. Stimulation of the M1 area in the precentral gyrus showed similar improvement [179]. Benefit was also found after targeting the dorsolateral PFC, after which both apathy and emotional processing improved [180]. Cognitive rehabilitation is beneficial in treating apathy in the healthy elderly, but no such benefits were observed in PD patients [181,182,183]. A pilot project showed slight benefits in the short term, but longitudinal data is not available [184].

## 7. Conclusions

Apathy is a marker of a distinct PD phenotype. It manifests itself during all stages of the disease, both in the prodromal stage and in advanced PD patients. Presence of apathy may fluctuate in individual patients, making assessment challenging. It is associated with earlier onset of axial symptoms, gait difficulties, motor complications, fatigue, and cognitive impairment. Patients with persistent apathy during follow-up may be at greater risk of developing these complications than those with incidental apathy. Whether severity of apathy plays a role is currently unclear. The underlying pathophysiology of apathy is complex, with different underlying neural networks resulting in separate apathy dimensions. These dimensions can be assessed through use of the LARS questionnaire. The LARS may prove a useful tool for tailoring therapy, as each dimension is associated with distinct neurotransmitter deficits. Additional studies are needed to elucidate how these different apathy dimensions present themselves in PD patients, how they evolve and respond to treatment. Thus far, tailored therapy is lacking but adequate DRT is recommended for all patients. Additional exercise interventions might be beneficial.

Future research should focus on follow-up of apathy in individual patients, monitoring evolution of presence, severity, and apathy dimensions during the disease course. Clinical trials focusing on treatment should take heed of apathy’s fluctuating nature, providing a long follow-up duration and multiple apathy assessments in time. A one-size-fits-all approach is to be avoided and future endeavours should consider underlying apathy dimensions as a guide of treatment choice and response.

## Figures and Tables

**Figure 1 brainsci-12-00923-f001:**
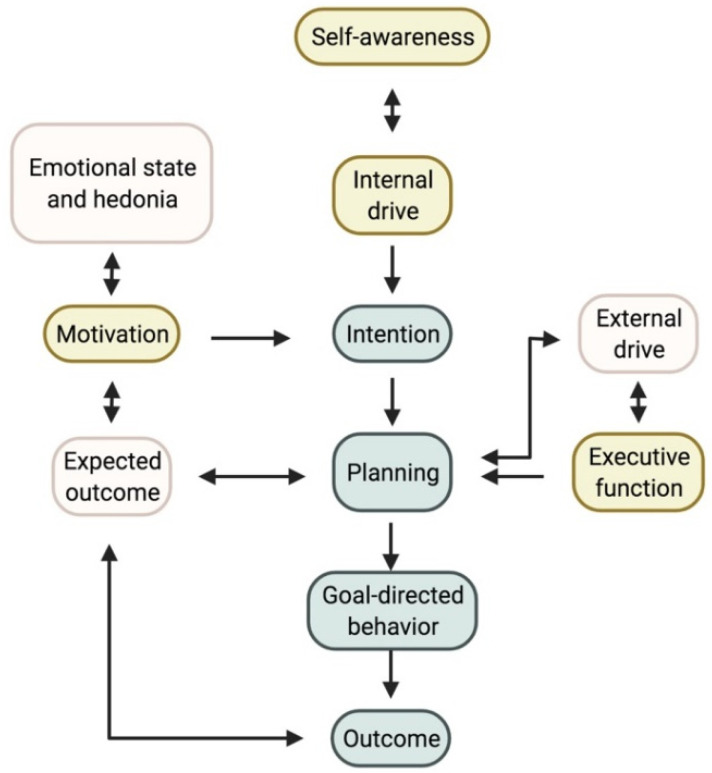
Proposed model for goal-directed behaviour. The three accepted subtypes and where they plug into the model are shown in yellow. Self-awareness, which is currently not yet considered a subtype, could interact with the internal drive and planning. Note on the left the importance of the hedonic state, which can affect those suffering from depression. Created with https://biorender.com/ (accessed on 27 June 2022).

**Figure 2 brainsci-12-00923-f002:**
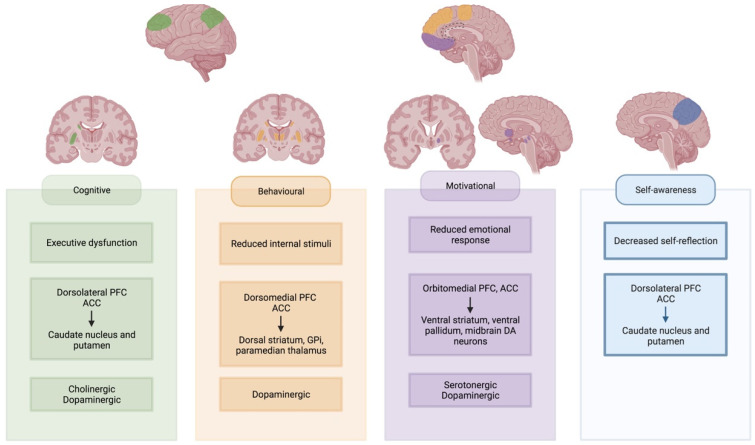
Neural networks underlying apathy subtypes. Involved cortical regions and basal ganglia regions are highlighted for each subtype. Created with https://biorender.com/ (accessed on 27 June 2022). Abbreviations: PFC: prefrontal cortex; ACC: anterior cingulate cortex; Gpi: internal globus pallidus; DA: dopaminergic.

## Data Availability

Not applicable.

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
