# Peer review of "Apathy in Parkinson’s Disease: Defining the Park Apathy Subtype"

_brainsci, 2022, doi:10.3390/brainsci12070923_

Round 1

Reviewer 1 Report

Authors:

I found your systematic literature review on the symptom of apathy in patients with Parkinson's disease to be especially well presented, thorough, and scientifically sound. You have presented an important and often overlooked aspect of basal ganglia dysfunction as it impacts the frontal cortex and cingulate gyrus functions. I hope that your article will impact the thinking of physicians on the devastating impact of this symptom in these patients. I have recommended acceptance of your article.

Author Response

We would like to thank the reviewer for their time and effort in reading and assessing our manuscript. We are very grateful for your kind feedback and recommendation of acceptance.

Reviewer 2 Report

Thanks for recommending me as a reviewer. In this review paper, authors were discuss the prevalence rates across the different disease stages and suggest screening tools for clinically relevant apathetic symptoms. Also, authores were approach the fundamental knowledge on the neural networks implicated in apathy in a practical manner and formulate recommendations on patient-tailored treatment. Authors were discuss the Park apathy phenotype in detail, shedding light on different clinical manifestations and implications for prognosis. If authors complete minor revisions, the quality of the study will be further improved.

1. The introduction section is well written. However, it is too short. It would be helpful to the reader if the authors describe the theoretical background of apathy in Parkinson's disease more specifically in the introduction section.

2. It will be helpful for readers to understand if the authors describe the conclusion section more specifically (e.g., implications for future research).

Author Response

Comment 1

“Thanks for recommending me as a reviewer. In this review paper, authors were discuss the prevalence rates across the different disease stages and suggest screening tools for clinically relevant apathetic symptoms. Also, authores were approach the fundamental knowledge on the neural networks implicated in apathy in a practical manner and formulate recommendations on patient-tailored treatment. Authors were discuss the Park apathy phenotype in detail, shedding light on different clinical manifestations and implications for prognosis. If authors complete minor revisions, the quality of the study will be further improved. The introduction section is well written. However, it is too short. It would be helpful to the reader if the authors describe the theoretical background of apathy in Parkinson's disease more specifically in the introduction section.”

We would like to thank the reviewer for their time and effort in reading our manuscript and providing their insightful feedback. We agree that the introduction is too concise. We provided a longer introduction, expanding on the background of apathy in PD. We have highlighted the changes in the manuscript. You may find them on page one, line 26-27 and line 32-45.

Comment 2

“It will be helpful for readers to understand if the authors describe the conclusion section more specifically (e.g., implications for future research).”

Thank you for pointing this out. We agree that the conclusion needs to be more precise and comprehensive for the reader. We have adjusted this in the manuscript on page 11, line 448-467.